# The Humanization and Maturation of an Anti-PrPc Antibody

**DOI:** 10.3390/bioengineering11030242

**Published:** 2024-02-29

**Authors:** Cheng Zhang, Fanlei Ran, Lei Du, Xiaohui Wang, Lei Liu, Jinming Liu, Quan Chen, Yang Cao, Lijun Bi, Haiying Hang

**Affiliations:** 1Key Laboratory of RNA Biology, Institute of Biophysics, Chinese Academy of Sciences, Beijing 100101, China; zhangc1mark@163.com (C.Z.); ranfanlei88@163.com (F.R.); 2University of Chinese Academy of Sciences, Beijing 100049, China; 3The State Key Laboratory of Membrane Biology, Institute of Zoology, Chinese Academy of Sciences, Beijing 100101, China; dul@ioz.ac.cn (L.D.); wangxhui@ioz.ac.cn (X.W.); liulei@ioz.ac.cn (L.L.); 4Beijing Institute for Stem Cell and Regenerative Medicine, Beijing 100101, China; 5The State Key Laboratory of Medicinal Chemical Biology, College of Life Sciences, Nankai University, Tianjin 300071, China; 18630864841@163.com (J.L.); chenq@nankai.edu.cn (Q.C.); 6Center of Growth, Metabolism and Aging, Key Laboratory of Bio-Resources and Eco-Environment of Ministry of Education, College of Life Sciences, Sichuan University, Chengdu 610064, China; cao@scu.edu.cn

**Keywords:** anti-PrPc antibody, affinity maturation, mammalian cell display, humanization, colorectal cancer

## Abstract

The cellular prion protein (PrPc) is a cell surface glycoprotein that is highly expressed in a variety of cancer tissues in addition to the nervous system, and its elevated expression is correlated to poor prognosis in many cancer patients. Our team previously found that patients with colorectal cancer (CRC) with high-level PrPc expression had significantly poorer survival than those with no or low-level PrPc expression. Mouse antibodies for PrPc inhibited tumor initiation and liver metastasis of PrPc-positive human CRC cells in mouse model experiments. PrPc is a candidate target for CRC therapy. In this study, we newly cloned a mouse anti-PrPc antibody (Clone 6) and humanized it, then affinity-matured this antibody using a CHO cell display with a peptide antigen and full-length PrPc, respectively. We obtained two humanized antibody clones with affinities toward a full-length PrPc of about 10- and 100-fold of that of the original antibody. The two humanized antibodies bound to the PrPc displayed significantly better on the cell surface than Clone 6. Used for Western blotting and immunohistochemistry, the humanized antibody with the highest affinity is superior to the two most frequently used commercial antibodies (8H4 and 3F4). The two new antibodies have the potential to be developed as useful reagents for PrPc detection and even therapeutic antibodies targeting PrPc-positive cancers.

## 1. Introduction

The cellular prion protein (PrPc) is a glycoprotein anchored on the surface of lipid rafts of the plasma membrane through the covalent binding of its C-terminus to the plasma membrane-buried GPI (Glycosylphosphatidylinositol). PrPc is expressed in the nervous system and immune cells under physiological conditions and is thought to play roles in nervous system development and immunity [1]. PrPc is thought to be involved in a variety of neurological diseases, and bovine spongiform encephalopathy (BSE) is caused by misfolded PrPc [2].

PrPc has also been found to be highly expressed in various cancers. Fan’s laboratory first found thd *PRNP* gene (encoding PrPc) was up-regulated in gastric cancer in 2002 and subsequently found that PrPc expression can enhance gastric cancer cell proliferation, migration and drug resistance [3,4,5]. The expression of *PRNP* is positively associated with tumor development, drug resistance and poor prognosis in colorectal cancer [6,7], pancreatic carcinoma [8,9,10], breast cancer [11,12], hepatocellular carcinoma [13], esophageal squamous cell carcinoma [14], glioma [15] and lung cancer [16,17]. Multiple studies show that the suppression of PrPc expression can reduce tumor cell viability. PrPc has the potential to be a novel cancer drug target [18,19,20].

Our team previously reported that anti-PrPc antibodies inhibited tumor initiation and liver metastasis of PrPc-positive human CRC cells in mouse model experiments [6]. In this study, we cloned a new mouse anti-PrPc antibody (Clone 6) which, in nude mice, can inhibit the migration of colorectal cancer stem cells (CCSC) originally derived from colorectal cancer patients and the growth of the CCSC. We intend to develop a therapeutic humanized anti-PrPc antibody for tumor treatment based on the mouse Clone 6 antibody.

In this study, we humanized the mouse antibody Clone 6 and obtained an antibody HAb 6. To increase the affinity of HAb 6 for the sensitive detection of PrPc in tissues and for sufficient efficacy of a therapeutic antibody candidate, a CHO display-based procedure previously developed by us was used. Briefly, we inserted antibody genes into CHO cells to be displayed as scFv on the membrane’s surface. The antibody mutation library was then constructed with AID (AID is able to deaminate deoxycytidines (dC) to deoxyuracils (dU), thus mutating the gene) [21,22,23,24]. We first matured the affinity of HAb 6 against the PrPc-35aa peptide, and then further matured its affinity against the full-length PrPc. This two-step maturation improved the affinity around 100-fold over HAb 6 against the full-length PrPc, and its K_D_ value was 2.03 × 10^−10^ M. This antibody binds well to both soluble PrPc and PrPc expressed in cells and fixed-tissue specimens. Compared to two commonly used commercial anti-PrPc antibodies (8H4 [8,9,25,26,27] and 3F4 [15,28,29,30,31]), our antibody demonstrates significantly stronger signals in Western blotting and immunohistochemistry.

## 2. Materials and Methods

### 2.1. Synthesis of Gene and Peptide and Expression of Protein

Antibody genes were codon optimized and synthesized by GenScript Biotech Corporation (Nanjing, China). Full-length antibody Clone 6 consists of a variable region from a mouse source and a constant region of human IgG1.

PrPc-35aa and FITC-PrPc-35aa sequences (in which PrPc-35aa conjugated FITC at its N-terminus) were synthesized by GenScript Biotech Corporation. The sequence of PrPc-35aa was GYMLGSAMSRPIIHFGSDYEDRYYRENMHRYPNQV. PrPc-Fc and antibodies were expressed in 293F cells and purified with protein A column (BBI Life Sciences Corporation, Shanghai, China).

### 2.2. Construction of Vectors

The primers for constructing vectors are listed in Appendix A.

pcDNA3.1-hygromycin-PrPc: *PRNP* was PCR-amplified with PrPc-NHE1F-primer and PrPc-XHO1R-primer and ligated into pcDNA3.1 plasmid at NHE1 and XHO1 digest sites.

pcDNA3.1-PrPc-Fc: *PRNP* sequence was obtained with PCR using primer PrPc-FcF1-primer and PrPc-FcR1-primers, and human Fc sequence was obtained with PCR using PrPc-FcF2-primer and PrPc-FcR2-primer. Finally, PrPc-Fc sequences were obtained by overlap and ligated into pcDNA3.1 plasmids using Hind3 and Xho1 enzyme cut sites.

pcDNA3.1-antibody-HC (heavy chain): The antibody mutants were obtained stepwise with point mutation using pcDNA3.1-HAb 6-HC as a template. A16D point mutations were generated with anti-PrPc-HC-A16DF-primer and PrPc-HC-A16DR-primer. Other mutations were also created in a similar manner.

pFRL-antibody-scFv-TM (transmembrane region sequence): The scFv form of HAb 6 and DV gene sequences contain the following elements in sequence: EcoR1 digest site, antibody signal peptide, antibody heavy chain variable region, (G4S)3 linker, antibody light chain variable region, HA tag, (G4S)3 linker, Xho1 digest site. ScFv form of HAb 6 gene sequence does not contain a transmembrane region sequence because pFRL plasmid has already carried a transmembrane region sequence and a stop codon after the Xho1 digestion site. The scFv form of HAb 6 and DV genes were ligated into the PFRT plasmid at the EcoR1 and Xho1 digest sites.

### 2.3. Cell Culture

293T and its derivative cells and P6C and its derivative cells were cultured in DMEM medium (HyClone, item no. SH30243.01) containing 10% FBS (HyClone). CHO/dhFr-cells and their derivatives were cultured in IMDM medium (HyClone, item no. SH30228.01) containing 10% FBS, 0.1 mM hypoxanthine and 0.016 mM thymidine (HT, Gibco, Waltham, MA, USA). 293F cells were cultured in SMM 293-TII medium (Sino Biological Inc., Beijing, China) in suspension. All cells were propagated at 37 °C, 5% CO_2_.

### 2.4. Transwell Assay

A single-cell suspension was counted and plated in the top chamber of a transwell (3422, Corning, New York, NY, USA) in serum-free medium. Both media in the top and bottom chambers included 5 μg/mL designated antibodies. After culturing for 24 h, cells that had migrated to the bottom chamber supplemented with 10% FBS (HyClone, Logan, UT, USA) were fixed and stained with Coomassie Blue. Migrated cells were counted in visual fields under microscope. Each sample was analyzed in triplicate and three independent experiments were performed. Irrelevant antibody is native IgG (bs-0297P, Bioss, Beijing, China), as a control antibody.

### 2.5. Organoid Formation

Human colorectal cancer (CRC) cells were developed by our team. After washing with phosphate-buffered saline (PBS), digested CRC cell suspensions were counted and incubated with growth factor-reduced, phenol-free Matrigel (Corning, 356231) and seeded in 48-well plates. The cells were covered with an organoid culture medium containing an N-2 (Gibco, 17502048) and B-27 supplement (Gibco, 17504044), GlutaMAX (Gibco, 35050061), N-acetylcysteine (A9165-5G, Sigma-Aldrich, Burlington, MA, USA), A-83-01 (Sigma-Aldrich, SML0788-25 MG) and SB202190 (Sigma-Aldrich, S7067-25 MG), and incubated at 37 °C at 95% humidity and 5% CO_2_. Single-cell suspensions were obtained using TrypLE Express protease (Gibco, 12605028) for 30 min, followed by filtration (40 μm). After washing with PBS, 500 cells were counted and incubated with 5 μg/mL designated antibodies in Matrigel and pictured at day 21.

### 2.6. Tumorigenesis in Orthotopic Xenograft Models

NOD/SCID mice were anesthetized with pentobarbital, and their ceca were exteriorized with laparotomy. A total of 50,000 DsRed-labeled P6C cells were injected into the cecal wall. Two weeks after tumor cell injection, the mice were randomly divided into 2 groups. Each group received IgG or anti-PrPc antibody injections at the concentration of 10 μg/kg body weight once a week, respectively. All the animals were monitored for 80 to 100 days or until tumor-associated death occurred. Primary tumors were evaluated with a whole-body fluorescence imaging system (In-Vivo FX PRO, Carestream, New York, NY, USA).

### 2.7. Transfection and Stable Cell Line Establishment

Cells were seeded for 24 h to achieve 70% confluence in a 6-well plate and transfected with 4 μg plasmids using 6 μg PEI MAX 40K. 293T-PrPc cell: 293T cells were transfected with pcDNA3.1-hygro-PrPc plasmid and cultured for 15 days at 100 μg/mL hygromycin. The cells highly expressing PrPc were enriched with flow sorting into 96-well plates. After the proliferation of the cells to sufficient numbers, the cell clone with the highest PrPc expression using flow assay was chosen as 293T-PrPc cell. P6C KO (knock-out): The P6C KO cell line was obtained by knocking out the P6C gene using crisp cas9. The sequences of gRNA were listed in Appendix A; P6C-OE (over-expression): P6C-OE cells were obtained with P6C transfection with pCEP4-hygro-PrPc followed by hygromycin drug screen.

### 2.8. Antibody Humanized Design

We used the BL21-CodonPlus strain to express full-length PrPc, immunized mice and obtained Clone 6 using the hybridoma technique [6].

The process of antibody humanization involves several steps. First, we utilized the AbRSA (version 1.0) [32] online tool to identify similar human germline V genes for the variable domains. Specifically, we found IGHV1-69-201 and IGKV4-101, which exhibited sequence similarities of 64.2% and 78.2%, respectively. Next, we applied the Chothia numbering scheme with AbRSA to delimit the framework regions (FRs) and complementarity-determining regions (CDRs). Subsequently, we replaced the FRs of the antibody sequence with those derived from the human V genes, resulting in the generation of the first humanized antibody. To gain further insights, we constructed a 3D structure model using Modeller (version 9v6) [33,34]. Manual counting was then employed to identify the contact residues between the CDRs and FRs, as well as the VL and VH interface. Contact residues were defined as those with a minimum atom distance between the residues of less than 5.5 Å. Based on the identified contact residues, we conducted back mutations on the first humanized antibody. We also compared the humanized antibody with the known human BCR sequence library to validate the designed sequences using Abalign (version 1.2.9) [35]. 

### 2.9. Affinity Maturation and Flow Cytometry

More details of affinity maturation and construction of pCI-Flp-2A-Cre and pCEP-mAID can be found in our previously published article [22]. All flow cytometry was performed on FACS AriaIII (BD).

When pFRL-antibody-scFv-TM plasmid and pCI-Flp-2A-Cre plasmid were co-transfected with CHO-PuroR-14, the recombinant enzymes Flpo and iCre expressed by pCI-Flp-2A-Cre plasmid were able to cleave the FRT and LOXP sequences in the genome and pFRL-antibody-scFv-TM plasmid so the antibody sequence released from the plasmid inserted into the cut site of the genome of CHO-PuroR-14. Cells that contained antibody genes were labeled with HA antibody anti-HA tag-APC (Columbia Biosciences, Frederick, MD, USA) and enriched with sorting. For the first round of maturation (from HAb 6 antibody), the cells displaying antibodies were transfected with pCEP-Neo-AID-plus and cultured under 1 mg/m LG418 for 10 days. The cells were labeled with FITC-PrPc-35aa and the anti-HA tag-APC antibody, and the cells that best bound FITC-PrPc-35aa and the anti-HA tag-APC antibody were enriched with flow cytometry sorting. The genomes of the sorted cells were purified with a genomic DNA purification system (Promega, Madison, WI, USA). The antibody genes were amplified using Hind3-kozak-SP-F and genome R primers, then sequenced. For the second round of maturation (from DV antibody), the cells displaying the DV antibody were first transfected with pCEP-Neo-AID-plus and cultured under 1 mg/mL G418 for 10 days. The cells were labeled with PrPc-Fc-FITC and the anti-HA tag-APC antibody, and the cells that best bound PrPc-Fc-FITC and the anti-HA tag-APC antibody were enriched with flow cytometry sorting. These cells were further transfected with pCEP-Bsd-AID and cultured under 10 µg/mL Blasticidin for 10 more days. Then the cells displaying mutant DV antibodies were labeled and enriched in the same way as above and the mutant DV antibody clones were sequenced.

### 2.10. Antibody Affinity Measurement

BLI experiments were performed with Octet Red96 (Forte Bio Octet, Menlo Park, CA, USA). The antibodies were captured with Anti-Human IgG Fc Capture (AHC) Biosensors (ForteBio Octet, Menlo Park, CA, USA) in an affinity assay using PrPc-35aa as the antigen, and PrPc-35aa were associated for 60 s and dissociated for 300 s. An affinity assay with PrPc-Fc as the antigen using Streptavidin (SA) Biosensors (Forte Bio Octet, Menlo Park, CA, USA) was used to capture biotinylated PrPc-Fc, antibody association for 300 s and dissociation for 900 s. KinExA assays were performed with the KinExA 4000 (Sapidyne Instruments Inc., Boise, ID, USA). Ten cell concentration half dilutions of 293T-PRPC and a constant concentration of antibody were incubated for 18 h. Then the cells were removed with centrifugation and the concentrations of free antibody in the 9 supernatants were detected, and the affinity between antibody and cells was calculated.

### 2.11. Western Blotting

Anti-PrPc antibodies (4AA-M, 8H4 or 3F4) were incubated at room temperature for 2 h (antibody concentration of 2 μg/mL) and anti-mouse IgG HRP antibody (PROMEGA, 1:5000 dilution) was incubated at room temperature for 1 h. Gray value was calculated using ImageJ software (version Fiji) from the National Institutes of Health (USA).

### 2.12. Immunohistochemistry

Bond RX (LEICA, Nussloch, German) was used for experiments. Serial sections of human cerebellar tissues were purchased from Bioaitech (Xi’an, China). Anti-PrPc antibodies were incubated at room temperature for 1 h. The OD values of IHC were calculated using the Halo analysis system.

## 3. Results

### 3.1. Murine Anti-PrPc Antibody Clone 6 Can Inhibit Migration and Metastasis

Our team generated a few anti-PrPc mouse hybridomas which inhibited the migration of colorectal cancer stem cells (CCSC) originally derived from colorectal cancer patients and the growth of the CCSC in nude mice [6]. Here we isolated a new antibody gene and sequenced it. We named this mouse antibody Clone 6. Clone 6, used below, was expressed with 293F and consists of a murine-derived variable region and a human IgG1 constant region. Clone 6 inhibited the migration and organoid formation of CCSC in vitro (Figure 1A,B); it was also able to greatly reduce the tumor growth from CCSC in nude mice (Figure 1C).

### 3.2. Humanization and Maturation of Clone 6 Antibody

We performed humanization and affinity maturation of antibody Clone 6, the process is summarized in Figure 2. The sequences of variable fragment of the individual antibodies in this process are in Figure 3A. The full-length Clone 6 antibody strongly bound to 293T cells overexpressing the full-length PrPc (293T-PrPc) (Figure 3B). We humanized the Clone 6 antibody and obtained three humanized antibodies: HAb 6, HAb 6b and HAb 6c (Appendix A). All three antibodies were able to bind to 293T-PrPc cells (Figure 3B). HAb 6 is the best binder among the three humanized antibodies and it can bind the cells even better than the original mouse antibody Clone 6. Therefore, we decided to mature HAb 6. Thirteen and six amino acid residues are different between HAb 6 and Clone 6 on their heavy chain variable region and light chain variable region, respectively.

We constructed an HAb 6 scFv gene and inserted it into a gene cassette with homologous recombination on a chromosome site in a CHO cell line previously developed in our laboratory [36]. The HAb 6 scFv was able to display on the cells (No-AID in Figure 3C). We generated a high-activity mouse AID (AID-plus) gene previously [22], and in this study, we transfected AID-plus into the cells displaying HAb 6 scFv. Random mutations on this gene were induced in the CHO cells with AID-plus by culturing the cells. PrPc-35aa, a 35 amino acid polypeptide in PrPc (127-161 amino acids in PrPc), was identified as the epitope of Clone 6 (unpublished data), and here FITC-PrPc-35aa was used to label the cells without HAb 6 (CHO-puro), the cells displaying parent HAb 6 (No-AID) and the cells displaying the pool of HAb 6 mutants (AID) (Figure 3C). Of note, the AID cells contained a small portion of cells that had higher FITC-PrPc-35aa binding ability than the majority of AID cells and all the No-AID cells. We enriched this portion of cells (10,000 cells within the rectangle) from 100 million cells with flow cytometry sorting and cultured the cells to 2 million. The enriched cells (Matured) contained two additional subpopulations which bound FITC-PrPc-35aa better than No-AID cells (Figure 3D). We further enriched the two subpopulations (sub1 and sub2) with flow sorting and cloned the antibody genes from the DNA isolated from the enriched cells. We sequenced 20 clones from sub1 and 27 clones from sub2. All the 20 sub1 clones contained a single mutation, A16D, and 25 of the sub2 clones contained a single mutation, A97V. The two other sub2 clones contained double mutation A97V-G107D and a single mutation, S76I, respectively. All four mutations were located in the heavy chain variable region.

To further characterize the highly enriched mutants A16D and A97V, full-length A16D, A97V and A16D-A97V antibodies were constructed, transiently displayed on CHO cells, and the cells were labeled with FITC-PrPc-35aa. The flow cytometric assay showed that A16D and A97V bound FITC-PrPc-35aa slightly better than HAb 6, and the binding of the A16D-A97V antibody to FITC-PrPc-35aa is obviously better than either A16D or A97V (Figure 3E). We synthesized full-length HAb 6 and A16D, A97V and A16D-A97V mutant antibodies and measured their affinities with full-length PrPc. All the mutants have higher affinities than the wild-type antibody (HAb 6: 1.24 × 10^−8^ M, A16D: 3.36 × 10^−9^ M and A97V: 5.15 × 10^−9^ M), and the affinity of the A16D-A97V antibody is 5.4 times that of the wild-type antibody HAb 6 (Table 1). We were curious how the original mouse antibody Clone 6, the humanized antibody HAb 6 and the A16D-A97V antibody would bind to the peptide PrPc-35aa, the soluble full-length PrPc and the full-length PrPc displayed on the 293T cells, respectively. To our surprise, the affinities of all the antibodies to PrPc-35aa were better than those to the soluble full-length PrPc and the full-length PrPc displayed on the 293T cells (Table 1). Actually, we could not obtain meaningful data on the affinities of Clone 6 and HAb 6 to the full-length PrPc displayed on the 293T cells because their binding to PrPc on 293T was too weak. One explanation is that the other regions beyond the 35aa peptide of PrPc influence the bindings of the antibodies to PrPc. Considering the application of these antibodies in therapy and diagnosis, antibodies that have high affinities to full-length PrPc are needed. The K_D_ values of the A16D-A97V antibody to the full-length PrPc in soluble form and displayed on cells are significantly smaller than those of Clone 6 and HAb 6 (1.13 × 10^−10^ M vs. 2.06 × 10^−9^ M and 1.01 × 10^−9^ M).

To further improve the affinity of the A16D-A97V antibody to the full-length PrPc by maturing the antibody towards the full-length PrPc instead of PrPc-35aa, we used FITC-conjugated PrPc-Fc to label cells. An A16D-A97V scFv gene was inserted in the gene cassette in CHO cells. In this maturation, two rounds of evolution were performed. The procedure for the first round is the same as described above for HAb 6 maturation. Afterward, the enriched cells were transfected with a plasmid-expressing mouse AID to induce mutations and then the cells that bound to the full-length PrPc were enriched with flow sorting. For the second round evolution, we intentionally used mouse AID instead of the high-activity version mouse AID-plus because the mutation type distributions of these two enzymes are not completely the same (our unpublished data) and we wanted to induce new mutations in the later rounds of evolution. After two rounds of evolution, the major cell subpopulation had higher binding ability than the cells displaying A16D-A97V scFv (Figure 4B). We sequenced 57 antibody gene clones derived from the cells after the second round of evolution (Table 2), and constructed full-length antibodies of these mutant genes, except the mutant A40T plus W insertion between W104 and G105. The two mutants A40T and W inserted between W104 and G105 were separately constructed and tested for their affinities (Table 3). BLI (Biolayer Interferometry) was used to measure the K_D_ values of these mutant antibodies except the mutants C96S and L4V (unable to express) and the mutant A40T plus F59L (unable to bind PrPc-Fc). To our surprise, the most enriched clones did not have significantly higher affinity than the parent clone (A16D-A97V). The only clone that binds to the full-length PrPc significantly better is the mutant containing four amino acids with APGK insertion between Q39 and A40 (Figure 4A).

We named the clone 4AA for the insertion of four amino acid residues and will use it in the rest of this article. We also analyzed the binding of 4AA to the full-length PrPc displayed on the 293T cells (293T-PrPc) with KinExA (Figure 4D). Its K_D_ value is 92.68 pM, and is about 10 times smaller than that of its parent clone A16D-A97V (Table 1). It is worthwhile to note that the two rounds of evolution against the full-length PrPc actually slightly decreased the affinity of the antibody to the PrPc-35aa peptide while increasing its affinity about 10 times to the full-length PrPc in either soluble form or displayed on the cell surface. The increase in the affinity of the antibody to the full-length PrPc could be important for its application in detection and therapy.

### 3.3. The Efficiency of the Inhibition of Tumor Cell Migration of the Matured Antibodies

We were curious whether the affinity-matured humanized antibodies were functionally stronger than HAb 6. 4AA inhibited the migration of the P6C CRC cells significantly better than Ab 6 as well as the DV antibodies, and the inhibition of the migration by DV is also significantly better than HAb 6 (Figure 4E), suggesting that each step of the affinity maturation increases their anti-migration functions.

### 3.4. The Specificity and Sensitivity of 4AA Antibody

To assess the specificity of 4AA, Western blotting was performed on the lysate of human CRC P6C cells [37] and *PRNP−/−* P6C cells. 4AA-m, the 4AA whose human IgG constant region was replaced with the mouse constant region, was used to label the blot. PrPc protein was positively detected in P6C cells, but not in the *PRNP* knockout P6C cells (Figure 5A). These data indicate that the 4AA-m antibody specifically binds to PrPc. Of note, the blot demonstrates multiple bands, suggesting that the PrPc has various post-translational modifications and/or cleavages. The 4AA-m antibody was also used to detect PrPc in the lysate of 293T and 293T cells that overexpress PrPc. PrPc protein was positively detected in both these cells (Figure 5B). However, 40 times more 293T cell lysate samples had to be loaded over that of the PrPc overexpressed cells so the PrPc protein of the 293T sample could be detected and compared with the sample of 293T-PrPc on the same blot, suggesting that 293T cells contain very low-level PrPc protein. These data on 293T and 293T-PrPc again confirm the specificity of the 4AA-m antibody to human PrPc.

Here we compared 4AA-m and the commercial antibodies 8H4 and 3F4 in a Western blotting analysis on 293T-PrPc cell lysate (Figure 5C). The 293T sample was not used in this comparison study because the expression level of PrPc was very low and an appropriate comparison of the sensitivity among these three antibodies could not be made. 4AA-m detected strong PrPc signals on both the 5 and 0.5 μg lysates, 8H4 can only detect a weak signal on the 5 μg lysate and 3F4 can also reveal strong and weak signals on the 5 and 0.5 μg lysates, respectively. We quantified the signals from the 5 μg lysate, and the signal detected with 4AA-m is nine and three times those with 8H4 and 3F4, respectively. This result suggests that 4AA-m is more sensitive than 8H4 and 3F4 in Western blotting analysis. It is worth noting that 4AA-m bound to the PrPc form with the lower apparent molecular weight (Figure 5C) slightly better than the form with the higher apparent molecular weight, while both 8H4 and 3F4 bound to the PrPc form with the higher apparent molecular weight significantly stronger.

There were conflicting reports on the association between the PrPc expression level in cancer tissues and the prognosis of cancer patients [25,29]. Tang et al. suspected that the difference in the abilities of the different antibody clones to specifically bind to PrPc in immunohistochemical analysis could be a reason. First, we examined the specificity of 4AA-m by performing immunohistochemistry on the fixed human CRC P6C and *PRNP−/−* P6C cells. 4AA-m stained P6C cells positively in contrast to the low background signals from *PRNP−/−* P6C cells while the two commercial anti-PrPc antibodies showed much smaller differences in signals between the P6C and *PRNP−/−* P6C cells (Figure 5D). The quantification of the densities of the specimens demonstrates that the signal ratios of P6C/P6C-KO when labeled with 4AA-m are significantly higher than those when labeled with 8H4 or 3F4, especially when the concentrations of the antibodies are lower. This result suggests that the specificity in immunohistochemistry of 4AA-m is superior to 3F4 and 8H4.

PrPc is highly expressed in human and other mammalian cerebellums [38,39]. We also performed immunohistochemistry on serial sections of the paraffin-embedded human cerebellum. 4AA-m revealed far stronger PrPc signals than 3F4 and 8H4 in the granular and molecular layers of the cerebellum, whose expression patterns are consistent with the results of previous studies [38], even at a concentration of 29- and 166-fold lower than those of 8H4 and 3F4, respectively (Figure 5E). The huge difference in sensitivity between our 4AA-m antibody and commercial 3F4 and 8H4 antibodies indicates that 4AA-m could be very useful in studying PrPc expression in human cells and human tissues.

## 4. Discussion

In this study, we cloned an anti-PrPc mouse antibody (Clone 6) and humanized and affinity-matured this mouse antibody, which generated two antibodies: DV and 4AA. The affinity of 4AA to the full-length PrPc is about 10 times higher than DV, and the K_D_ values of the 4AA to the soluble full-length PrPc and the full-length PrPc displayed on cells are 2.03 × 10^−10^ M and 9.27 × 10^−11^ M.

Before the humanization and affinity maturation, the mouse antibody Clone 6 was exhibited to be able to inhibit the migration, organoid growth and tumor growth in nude mice of human CRC cells (Figure 1A–C). These results ensured the high probability we would acquire a humanized anti-PrPc antibody with a potentially therapeutic function. The K_D_ value of Clone 6 to the soluble PrPc is 1.24 × 10^−8^ M, 10.1 and 103-fold of its humanized counterparts DV and 4AA, respectively. After humanization and maturation, the abilities of migration inhibition of human CRC cells by these antibodies were compared, and the inhibition strengths of DV and 4AA were significantly stronger than Clone 6, suggesting that the improved affinities of DV and 4AA can transfer to higher anti-tumor function. Thus, the two humanized antibodies can be potentially developed into therapeutic antibodies against CRC.

In this study, we used AID-plus and AID to mutate the anti-PrPc antibody. The mutation efficiency of the AID and AID-plus engineered by our group is much higher than the original wild-type AID [22]. This high mutation efficiency guarantees that the affinity maturation based on CHO display becomes a practical and efficient method compared to the other affinity maturation techniques. Compared to the other affinity maturation methods (e.g., error-prone PCR mutation in phage/yeast screening), this technological platform has several useful unique features. First, it is very easy to construct mutant libraries by simply growing CHO cells. Second, CHO cells are the most frequently used host cells to produce therapeutic antibodies [40] and other therapeutic proteins [41]. The matured antibodies and proteins via this maturation platform are more readily transferred to production without potential problems involving post-translational modifications and production yield. The mouse Clone 6 antibody was originally mapped to bind to a 35aa peptide (AA127-161). Therefore, its humanized antibody HAb 6 was first matured against the FITC conjugated peptide PrPc-35aa-FITC. The maturation using this peptide generated DV. When the K_D_ values of DV to PrPc-35aa, soluble full-length PrPc and full-length PrPc displayed on human 293T cells were compared, its value to PrPc-35aa is about 10- and 5-fold of those to soluble full-length PrPc and full-length PrPc displayed on cells, respectively (Table 1). These large differences in affinity alerted us since the full-length PrPc would be the real target in either therapy or diagnosis. We further matured DV by using full-length PrPc-Fc and obtained 4AA. The K_D_ values of 4AA to soluble full-length PrPc and full-length PrPc displayed on cells decreased more than 10 times those of DV while its K_D_ value to PrPc-35aa increased 2.8-fold from that of DV. These data indicate that other amino acid residues in addition to PrPc-35aa are important for the binding of 4AA to full-length PrPc.

The antibody 4AA-m can specifically recognize PrPc in Western blotting and IHC. Compared to the frequently used commercial PrPc antibodies 3F4 and 8H4, 4AA-m was far more sensitive in both assays (Figure 5). Thus, 4AA-m is very likely to be widely useful in future studies of PrPc, and even in diagnoses.

The role of PrPc in tumorigenesis is controversial. For example, an early study by Zhou et al. reported that the overexpression of PrPc is predictive of poor prognosis in gastric cancer [29]. However, a later study by Tang et al. demonstrated that negative PrPc expression was associated with poor overall survival [25]. Tang et al. suggested that the opposite results between the two studies might have resulted from the difference between these two commercial antibodies used in IHC in the tissue samples from the patients; Tang et al. used the 8H4 clone, while Zhou et al. used the 3F4 in their studies. Although Zhou did not specifically point out that they used 3F4 in this article, we received an email confirmation that they did use 3F4 from one of the senior authors of this article. In our current study, we paid special attention to the specificities of these two commercial antibodies and 4AA. In the IHC of the CRC cancer cell P6C and P6C-PrPc KO (clone 209 in Figure 5D), both 3F4 and 8H4 stained P6C cells better than P6C-PrPc KO cells (Figure 5D). The two antibodies also stained the PrPc-overexpressing 293T cells better than 293T cells (Figure 5C). They can also specifically stain sections of the human cerebellum (Figure 5E). Compared to 3F4 and 8H4 in these assays, 4AA produced far stronger signals. In the IHC quantitative analysis using P6C and P6C-PrPc KO cells, the sensitivity and specificity are clearly superior to 3F4 and 8H4 (Figure 5D). At this point, the controversial issue has not been completely resolved. One of the tests that will help to resolve the issue is to test these antibodies on tissues that normally express PrPc and the tissues with the deletion of PrPc.

## 5. Conclusions

In this paper, we obtained a new murine anti-PrPc antibody (Clone 6) with the ability to inhibit tumor development, which inhibits CCSC migration and organoid formation in vitro and significantly reduces tumor growth in nude mice in vivo. We humanized Clone 6 and obtained two high-affinity antibodies, DV and 4AA, using epitope peptides and full-length proteins as ligands by the CHO cell display system, respectively. They showed high potential for both diagnosis and treatment of PrPc targets. DV and 4AA retained their tumor inhibitory function and were significantly more inhibitory than Clone 6. Recombinant 4AA-m possessed higher sensitivity than commercial antibodies in detecting PrPc. Furthermore, the optimized method of using antigens in different forms can effectively improve the affinity of antibodies against natural antigens, providing a new possible strategy for antibody affinity maturation.

## Figures and Tables

**Figure 1 bioengineering-11-00242-f001:**
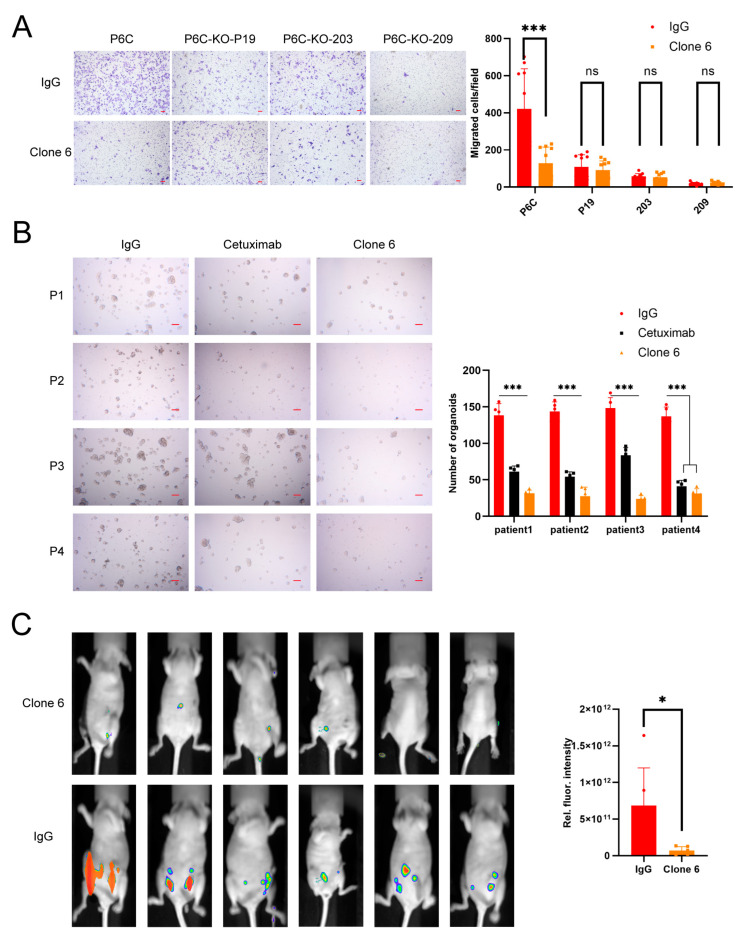
Murine anti-PrPc antibody Clone 6 can inhibit the migration and metastasis of colorectal cancer cells. (**A**) Clone 6 (5 µg/mL) inhibited migration of colorectal cancer cell P6C in transwell assay but not the PrPc-knockout P6C cells (P6C-KO-P19, P6C-KO-203 and P6C-KO-209). The information on the PrPc-knockout P6C cells is in Appendix A. Irrelevant IgG was used as control. On the left are representative pictures (scale bar, 100 µm), and on the right are statistic results (ns, not significant; *, *p* < 0.05; ***, *p* < 0.001). (**B**) Clone 6 (5 µg/mL) inhibits the formation of organoids derived from colorectal cancer cells (500 cells) originally isolated from 4 colorectal cancer patients. The left panel shows representative photographs (scale bars, 200 μm). Both cetuximab (positive control) and Clone 6 inhibited the organoid formation. Irrelevant IgG was used as a negative control. The right panel is the statistical results of 5 dishes (500 cells per dish). The inhibition by Clone 6 in 3 of the 4 patient samples is far more significant than the inhibition by cetuximab (*p* < 0.001). (**C**) The left shows representative whole-body fluorescence imaging of growing tumors from 50,000 P6C colorectal cancer stem cells in each mouse treated with 10 μg/kg Clone 6 or irrelevant IgG. On the right are the statistical results.

**Figure 2 bioengineering-11-00242-f002:**
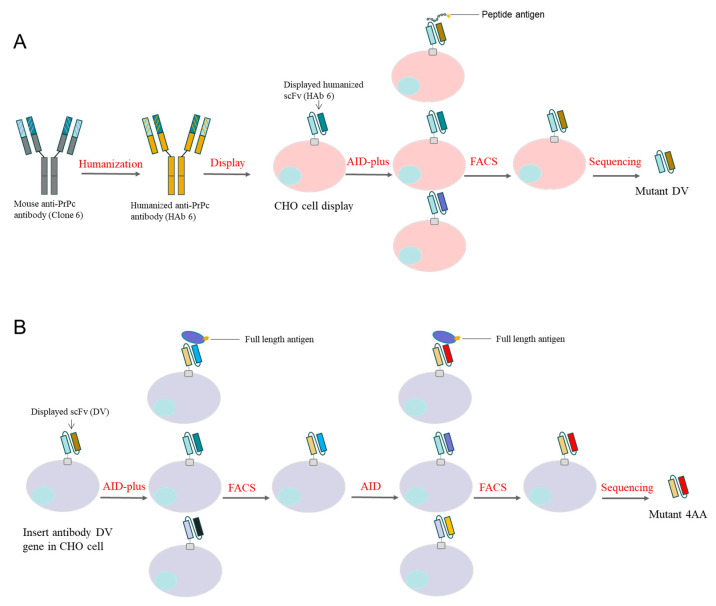
Schematic representation of affinity maturation in the study. (**A**) Mouse antibody Clone 6 was humanized to HAb 6. The HAb 6 antibody gene was inserted into the CHO cell line and displayed as scFv on the surface of the cell membrane. Antibody mutant libraries were obtained by growing the CHO cells in which HAb 6 mutants were induced by AID-plus, and then the cells displaying antibody mutants with high affinity were enriched with flow cytometry sorting using PrPc peptide antigen as a labeling ligand. The affinity-enhanced mutant DV was obtained with sequencing. (**B**) The DV antibody gene was reinserted into the CHO cell line. Two rounds of artificial evolution were performed sequentially on the antibody mutant libraries created with AID-plus and AID, respectively, using the full-length protein PrPc as a labeling ligand. Mutant 4AA with further improved affinity was obtained with sequencing.

**Figure 3 bioengineering-11-00242-f003:**
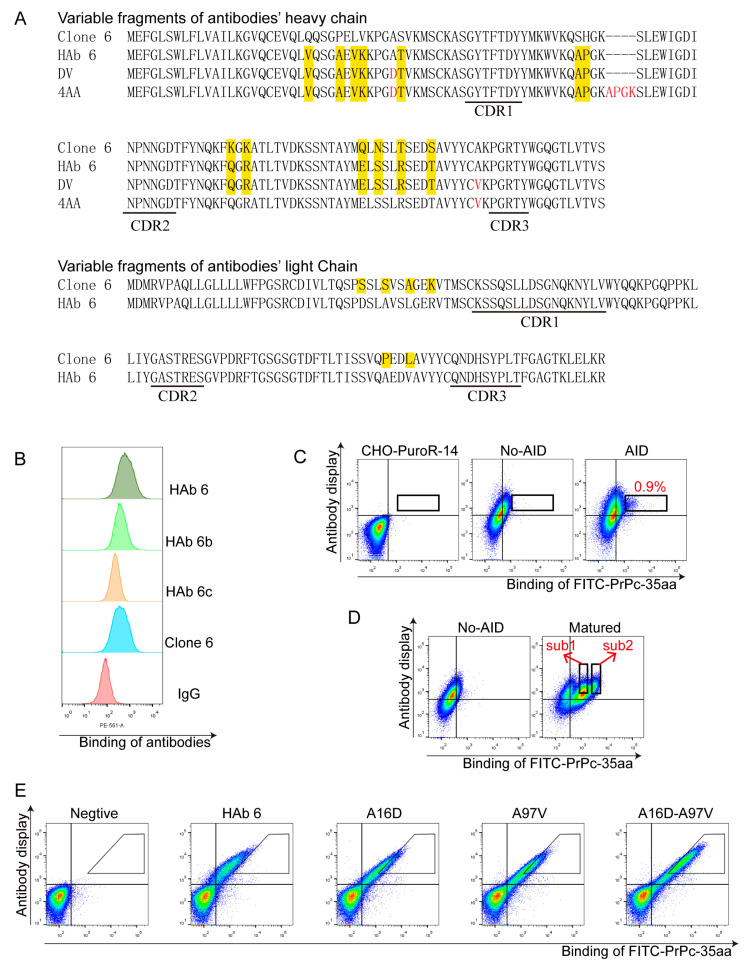
Humanization and affinity maturation of Clone 6. (**A**) Sequences of variable fragment of antibodies. HAb 6 is the antibody humanized from Clone 6. DV and 4AA are the antibodies affinity-matured from HAb 6. The amino acid residues marked with the yellow background are different from those in the same locations of Clone 6. Red letters representing the amino acid residues of DV and 4AA are different from those of Clone 6 and HAb 6. The “—” stands for no corresponding amino acids. The deduced amino acids with CDRs are marked. (**B**) Flow cytometry data comparing Clone 6 and 3 humanized anti-PrPc antibodies. 293T-PrPc were labeled with Clone 6 and 3 humanized anti-PrPc antibodies. HAb 6 bound to 293T-PrPc better than all the other antibodies. Antibody concentration was 0.5 μg/mL. (**C**) Generation of mutant pool and sorting strategy. CHO-PuroR-14 are the original cells without the HAb 6 scFv gene inserted. No-AID are the parent cells with HAb 6 scFv gene inserted. AID are mutant cells generated by transfecting No-AID cells with AID-plus. The cells in the rectangle in AID which bound more FITC-PrPc-35aa at similar antibody display levels were enriched with flow sorting. (**D**) Affinity-matured cells were those enriched from AID cells. The matured cells are composed of 3 subpopulations, and sub1 and sub2 displayed antibody mutants with higher binding abilities. (**E**) Flow cytometry of HAb 6 mutants (A16D, A97V and A16DA97V) in full length labeled with PrPc-35aa-FITC. A16D and A97V were identified by sequencing mutant antibody clones isolated from sub1 and sub2 cells. Cells displaying A16D, A97V and A16D-A97V mutant antibodies bound to FITC-PrPc-35aa better than those displaying HAb 6. The concentration of FITC-PrPc-35aa is 1 μg/mL in (**C**–**E**).

**Figure 4 bioengineering-11-00242-f004:**
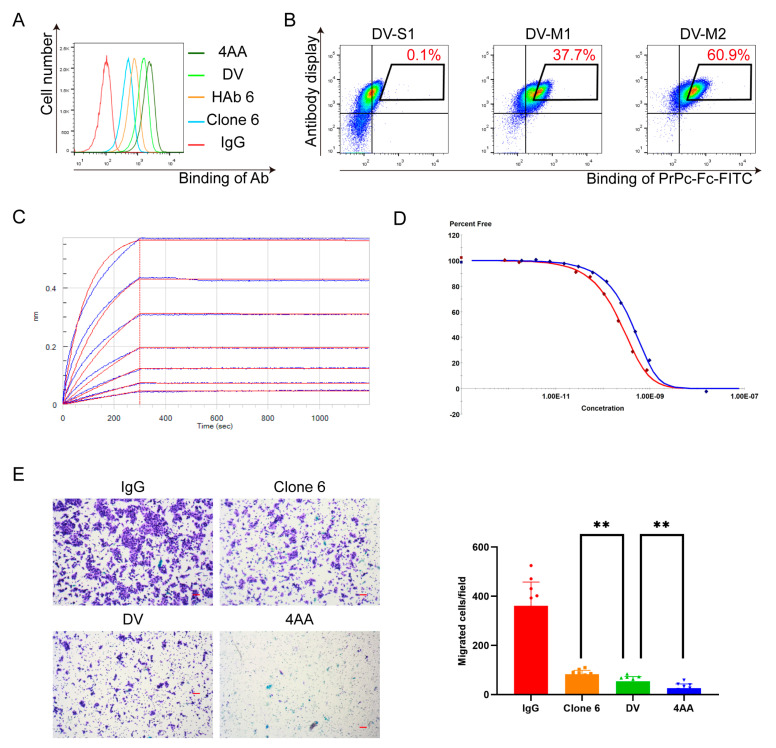
Directed evolution of DV antibody and identification of 4AA. (**A**) Binding abilities of antibodies to 293T-PrPc measured with flow cytometry. Antibodies were expressed as full-length antibodies in 293F. Antibody concentration was 0.5 μg/mL. (**B**) Directed evolution of DV antibody by labeling with PrPc-Fc-FITC. DV-S1 are cells that expressed antibody mutant A16D-A97V in scFv form. AID-plus was transfected into DV-S1 to induce DV mutations and the cells displaying mutants with higher binding abilities (DV-M1) were enriched. DV-M1 cells were transfected with AID, then the cells displaying mutated DV-M1 antibodies with higher binding abilities to PrPc-Fc (DV-M2) were further enriched. The concentration of PrPc-Fc-FITC is 2 μg/mL. (**C**) The BLI assay results between PrPc-Fc and the optimized anti-PrPc antibody 4AA are shown here. 4AA was expressed as full-length antibody in 293F cells. The affinity between 4AA and PrPc-Fc was 2.03 × 10^−10^ M. (**D**) KinExA assay result between 293T-PrPc and DV (blue) and 4AA (red). The affinities of DV and 4AA to 293T-PrPc are 1.01 nM and 92.68 pM, respectively. (**E**) Evaluation of inhibition of tumor migration by various antibodies. 4AA inhibited P6C migration better than DV, and DV was better than Clone 6 (**, *p* < 0.01).

**Figure 5 bioengineering-11-00242-f005:**
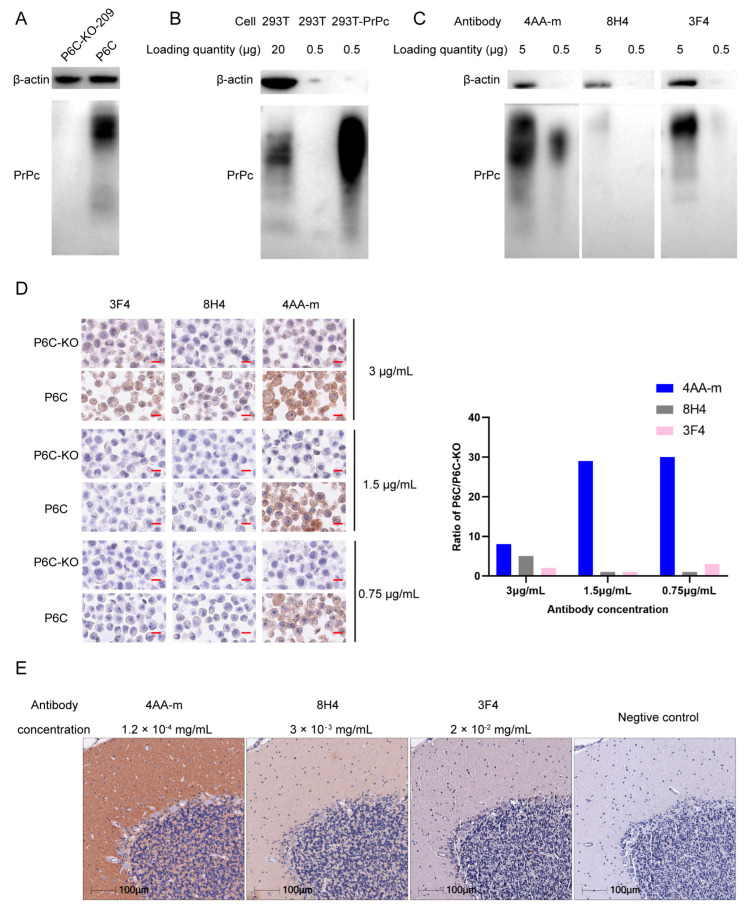
The specificity and sensitivity of 4AA antibody. (**A**) Western blotting on the lysate of PrPc-KO-209 and P6C cells using 4AA-m. A total of 10 μg lysate was loaded. (**B**) Western blotting on the lysate of 293T-PrPc and 293T cells using 4AA-m. A total of 20 or 0.5 μg 293T lysate was loaded, and 0.5 μg 293T-PrPc lysate was loaded. (**C**) Western blotting on the lysate (10 or 0.5 μg) of 293T-PrPc cells using 4AA-m, 8H4 or 3F4. (**D**) IHC on P6C or PrPc-KO-209 cells using 4AA-m, 8H4 or 3F4 at designated concentrations (**left**) and quantification of ratio of OD of P6C/PrPc-KO-209 (**right**). (scale bar, 15 µm) (**E**) IHC on human cerebellum using 4AA-m, 8H4 and3F4 at designated concentrations.

**Table 1 bioengineering-11-00242-t001:** Affinity results between antibodies and different forms of PrPc.

	Affinity to Different Forms of PrPc K_D_ (M)		
Antibody	PrPc-35aa	PrPc-Fc	293T-PrPc
Clone 6	3.14 × 10^−10^	2.09 × 10^−8^	N.D.
HAb 6	2.92 × 10^−9^	1.24 × 10^−8^	N.D.
A16D-A97V	1.13 × 10^−10^	2.06 × 10^−9^	1.01 × 10^−9^
4AA	2.81 × 10^−10^	2.03 × 10^−10^	9.27 × 10^−11^

The affinities of anti-PrPc antibodies to PrPc-35aa and PrPc-FC were measured using BLI. The affinities of anti-PrPc antibodies to 293T-PrPc were measured using KinExA. All antibodies were expressed as full-length human IgG1 in 293F. N.D.—not detected.

**Table 2 bioengineering-11-00242-t002:** Sequence of antibody genes cloned from DV-M2.

Mutations	Count
A40T	27
Q39K plus S76I	10
W insertion between W104 and G105	4
Q39K	2
S76I	2
A40T plus W insertion between W104 and G105	2
APGK insertion between Q39 and A40	2
Wild type	2
A40T plus S76I	1
C96S	1
S76N	1
L4V	1
A40T F59L	1
S76N plus S58N	1

**Table 3 bioengineering-11-00242-t003:** Affinities of Antibody mutants to PrPc-Fc.

Antibody Mutants	K_D_ (M)	Kon (1/Ms)	koff (1/s)
Parent clone (A16D-A97V)	2.06 × 10^−9^	1.17 × 10^4^	2.42 × 10^−5^
4AA	2.03 × 10^−10^	1.12 × 10^4^	2.27 × 10^−6^
A40T	1.33 × 10^−9^	1.11 × 10^4^	1.48 × 10^−5^
A40T plus S76I	2.47 × 10^−9^	9.09 × 10^3^	2.25 × 10^−5^
Q39K	3.14 × 10^−9^	1.08 × 10^4^	3.40 × 10^−5^
Q39K plus S76I	3.55 × 10^−9^	1.05 × 10^4^	3.72 × 10^−5^
S76I	4.86 × 10^−9^	1.29 × 10^4^	6.27 × 10^−5^
S76N plus S58N	6.91 × 10^−9^	1.20 × 10^4^	8.29 × 10^−5^
S76N	7.76 × 10^−9^	1.18 × 10^4^	9.18 × 10^−5^
W insertion between W104 and G105	3.17 × 10^−8^	4.73 × 10^3^	1.50 × 10^−4^

All antibodies were expressed as human IgG1. Affinities were measured using BLI.

## Data Availability

The raw data supporting the conclusion of this article will be made available by the authors, without undue reservation.

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
