# Peer review of "The Humanization and Maturation of an Anti-PrPc Antibody"

_bioengineering, 2024, doi:10.3390/bioengineering11030242_

Round 1
Reviewer 1 Report
Comments and Suggestions for Authors
bioengineering-2832572 review
This study by Zhang et al utilized AID-mediated mutagenesis for antibody affinity maturation and further applied the isolated mutants for in vitro cellular tests. Overall, the study is complete and worth for reporting. Please see following for my comments – hopefully to make the presentation clearer and improve the significance. I also point some language issues.
Major:
1. I would like to suggest adding a schematic cartoon as Fig 1 to summarize the whole procedure of humanization and affinity of current study.
2. In the last paragraph of Introduction, please add description of AID-mediated mutagenesis, as the key / unique method used for affinity maturation in this study.
3. Fig 1ABC, please plot dots on top of bar graph – it now becomes standard for data presentation. Apply to Fig 3E as well.
4. Section 3.2, line 227, please describe in Result how Clone 6 was humanized?
5. Fig 2BC, the data seem largely redundant, suggest removing panel B, or merging it with panel C.
6. Fig 2BC, add important info on the concentrations of IgG used in Result text and/or figure legend. The same for Fig 3A.
7. Fig 2DE, add important info on the concentrations of antigens used in Result text and/or figure legend. The same for Fig 3B.
8. Line 309-311, what is the AID-in and its difference from AID-plus? It is not clear at all.
9. In Discussion, please add comments on AID-in and AID-plus, and comparing this method with other affinity maturation approaches.
Minor:
Line 24, “with different antigens”, it will be better to briefly describe / specify these antigens
Line 36, “to the extracellular surface” -> on surface
Line 37, “covalently binding” -> “covalent binding of”
Line 39, 40, remove “is believed to”, “thought to be”
Line 49, “show” -> “showed”
Line 61, “KD” -> subscribe D. Please apply to all “KD” in manuscript
Line 62, “2.03E-10” -> 2.03 x 10-10. Apply to entire manuscript
Line 62, remove “can”
Section 2.2, remove bullet points
Line 104, “CO2”, subscribe 2
Section 2.7, one paragraph will be fine
Line 224, 254, “full-length” -> human IgG1 in the constant? If so, please add the info in Results
Line 234, “a CHO cell line previously developed in our laboratory.” Need citation(s) or description.
Table 2 and 3, I feel it’s better to combine them
Table 3, “Kdis” -> koff?
Fig 3A, x-axis label -> binding of Ab
Line 408 “@g”?
Line 420, “was experimentally shown to be able to” -> exhibited to
Line 424, remove “and”
Comments on the Quality of English Language
See above
Reviewer 2 Report
Comments and Suggestions for Authors
The manuscript may be accepted for the publication.
Author Response
We are pleased that the reviewer suggests acceptation of our manuscript for publication.
Reviewer 3 Report
Comments and Suggestions for Authors
It is a very interesting article. The authors was carrying out two step humanization of mouse antibody Clone 6 in order to obtain an humanized anti-PrPc antibody HAb 6 antibody for tumor treatment.
Suggestion:
1. The author mentioned the antibody Clone 6 was reported in this article "CD44-positive cancer stem cells 525 expressing cellular prion protein contribute to metastatic capacity in colorectal cancer".
- Can you briefly to describe how did you obtain this antibody in the methodology section?
2. In Figure 2A, possible to improve the deduced amino acids with CDRs region?
Author Response
Dear Reviewer:
We are grateful to you and the reviewers for their valuable comments and suggestions which help improve the quality of our manuscript titled “The Humanization and Maturation of an Anti-PrPc Antibody”. We have tried hard to address all the questions and concerns raised by you and the reviewers on our manuscript. The added and modified contents in the revised manuscript have been marked in red. I hope that our answers to the questions and the revised manuscript meet your requirements.
Comments 1: The author mentioned the antibody Clone 6 was reported in this article "CD44-positive cancer stem cells 525 expressing cellular prion protein contribute to metastatic capacity in colorectal cancer".
- Can you briefly to describe how did you obtain this antibody in the methodology section?
Response 1: We have added the description to the revised manuscript in 2.8. Antibody Humanized Design follows:
We used the BL21-CodonPlus strain to express full length PrPc, immunized mice, and obtained Clone 6 by the hybridoma technique [6].
Comments 2: In Figure 2A, possible to improve the deduced amino acids with CDRs region?
Response 2: We have modified this in Figure 3A (it used to be Figure 2A).
Round 2
Reviewer 1 Report
Comments and Suggestions for Authors
all my comments are well addressed.